# Fossil brains provide evidence of underwater feeding in early seals

George A. Lyras[1], Lars Werdelin [2], Bartholomeus G. M. van der Geer[3] & Alexandra A. E. van der Geer [4,5✉]

Pinnipeds (seals and related species) use their whiskers to explore their environment and locate their prey. Today they live mostly in marine habitats and are adapted for a highly specialised amphibious lifestyle with their flippers for locomotion and a hydrodynamically streamlined body. The earliest pinnipeds, however, lived on land and in freshwater habitats, much like mustelids today. Here we reconstruct the underwater foraging behaviour of one of these earliest pinnipeds (*Potamotherium*), focusing in particular on how it used its whiskers (vibrissae). For this purpose, we analyse the coronal gyrus of the brain of 7 fossil and 31 extant carnivorans. This region receives somatosensory input from the head. Our results show that the reliance on whiskers in modern pinnipeds is an ancestral feature that favoured survival of stem pinnipeds in marine habitats. This study provides insights into an impressive ecological transition in carnivoran evolution: from terrestrial to amphibious marine species. Adaptations for underwater foraging were crucial for this transition.

[1] Faculty of Geology and Geoenvironment, Department of Historical Geology-Palaeontology, National and Kapodistrian University of Athens, 15784 Zografos, Greece. [2] Department of Palaeobiology, Swedish Museum of Natural History, SE-10405 Stockholm, Sweden. [3] Casa del Vento, 50060 FI Santa Brigida, Italy. [4] Vertebrate Evolution, Development and Ecology, Naturalis Biodiversity Center, 2333 RA Leiden, the Netherlands. [5] Institute of Biology, Leiden University, 2311 EZ Leiden, the Netherlands. ✉email: alexandra.vandergeer@naturalis.nl

The shift from a terrestrial to an aquatic lifestyle in the evolution of the mammalian order Pinnipedia (seals, sea lions and walruses) is an impressive ecological transition in carnivoran evolution[1]. Adaptations for underwater feeding and foraging were crucial for this transition[2]. While the feeding strategy of fossil pinnipeds can be inferred from their teeth and skeleton[2,3], their foraging behaviour is more challenging to reconstruct. Modern pinnipeds use their whiskers (vibrissae) to explore their environment by detecting vibrations in the water, e.g., refs. [4,5]. This includes hydrodynamic prey sensing, as observed in harbour seals (*Phoca vitulina*), elephant seals (*Mirounga angustirostris*) and ringed seals (*Pusa hispida*)[6,7]. We do not know, however, when this behaviour first appeared during pinniped evolution. The most basal stem pinnipeds differed considerably from their modern counterparts. They were otter-like animals occupying freshwater environments[8]. These stem pinnipeds occupied the ecological gap between semiaquatic freshwater and semiaquatic marine species. Among stem pinnipeds *Puijila*, arguably the most basal form, was the least specialised for swimming[8]. *Potamotherium* had webbed feet and was adapted to freshwater habitats[9]. Both *Puijila* and *Potamotherium* were otter-like in appearance. *Enaliarctos*, a more derived stem pinniped, occupied marine habitats, and had a streamlined body, a reduced tail, and limbs that were highly modified to form flippers[10]. During this transition, the underwater foraging strategy may have changed from otter-like behaviour to that which we see in pinnipeds today.

Living freshwater species, not directly related to pinnipeds, can be used as analogues of what that otter-like foraging behaviour of stem pinnipeds could have looked like. Modern otters follow two principal foraging ecologies. Some otters, e.g., the Eurasian otter (*Lutra lutra*), use a piscivorous mouth-oriented foraging behaviour, whereas others, e.g., the African clawless otter (*Aonyx capensis*) favour an invertebrate hand-oriented mode of predation[11–13]. Similarly, the marsh mongoose (*Atilax paludinosus*) forages by feeling for prey with its forepaws[14], whereas the otter civet (*Cynogale bennettii*), uses its whiskers to locate food on the stony bottom of rivers[15]. Thus, modern analogues indicate that early pinnipeds could have used either a whisker- or a hand-foraging behaviour.

Preserved remains of whiskers or associated soft tissues are unknown for stem pinnipeds. There are, however, other indicators of the importance of whiskers. Whisker-foraging mammals have thicker infraorbital nerves and thus a wider infraorbital foramen, through which these nerves pass[16]. Stem pinnipeds have enlarged infraorbital foramina as well[8], which indicates that whiskers may have played an important role in their behaviour. However, the size of the infraorbital foramen alone cannot be used to predict whisker sensitivity, because its size depends not only on the number of whiskers, but also on the innervation of the individual whiskers[17,18]. Simply stated, a species with fewer but strongly innervated whiskers may have a similarly-sized infraorbital foramen as a species with more but minimally innervated whiskers. An alternative way to address facial sensory abilities of fossil pinnipeds is palaeoneurology, the study of fossil brains. The endocranial casts of carnivorans preserve many details of their external brain anatomy[19]. Thus, the endocranial anatomy of fossil pinnipeds and their extant relatives can inform us about the patterns of ridges (gyri) and fissures (sulci) of their brain. Electrophysiological brain mapping data on northern fur seals (*Callorhinus ursinus*)[20], sea lions (*Zalophus californianus*)[21] as well as on canids, felids and procyonids[22–26], demonstrate that one particular gyrus, known as the coronal gyrus, receives somatosensory projections from the head, and particularly from the vibrissae. Therefore, the relative size of the coronal gyrus can inform us about the somatic sensory specialisations of fossil pinnipeds.

We, therefore, used external brain anatomy as preserved in endocranial casts to reconstruct the foraging ecology of pinnipeds at the time of the transition from terrestrial to marine habitats. To do so, we investigated the coronal gyrus of *Potamotherium*, the only freshwater stem pinniped for which endocasts are known[9,11,27–31] and compared it with that of fossil and living carnivorans . This is crucial for constructing an accurate picture of an important transition in carnivoran evolution.

We found that the coronal gyrus remained narrow in most terrestrial carnivorans. In contrast, the coronal gyrus expanded disproportionately in semi-aquatic carnivorans that use their whiskers for exploring their environment, while it did not expand in carnivorans that use their hands in foraging. *Potamotherium* shows an increased size of the coronal gyrus, which provides further evidence that it was a whisker specialist. We postulate that the increased tactile performance of mystacial vibrissae of modern pinnipeds was already present around the beginning of the transition from a terrestrial to an aquatic lifestyle and facilitated their transition to an amphibious lifestyle.

## Results

**Fossil carnivorans**. The brain of the stem pinniped *Potamotherium* is pear-shaped (Fig. 1 and Supplementary Fig. 1a, b). The sigmoid gyri (Fig. 1c) are located rostrally. They are narrow, without a clear cruciate sulcus. The coronal gyrus is much wider than that of terrestrial carnivorans of similar size (see below) and is subdivided by a shallow secondary sulcus (Fig. 1c). Due to the expansion of the coronal gyrus, the anterior limb of the suprasylvian sulcus is shifted ventrally. Nevertheless, the anterior ectosylvian gyrus is as wide as the posterior ectosylvian gyrus. At the apex of the suprasylvian arc there is a short, medially directed spur.

*Enaliarctos*, another stem pinniped (Fig. 1b), has an expanded coronal gyrus, with a longer secondary sulcus. The gyrus is more rostrally located than in *Potamotherium* (Supplementary Fig. 1c, d). The rostral end of its brain is relatively high and flat and the cerebral hemispheres are relatively flattened dorsally.

In *Pinnarctidion* the coronal sulcus is very broad, nearly vertically oriented and overlaps the anterior arm of the ectosylvian gyrus. The posterior ectosylvian gyrus is broader than the anterior ectosylvian gyrus. The sigmoid gyri are located extremely rostrally (Supplementary Fig. 1e, f). Just as in *Enaliarctos*, the rostral end of the brain is relatively high and caudally compressed. The cerebral hemispheres are relatively flattened dorsally.

*Mionictis* has an expanded coronal gyrus, with a secondary sulcus (Supplementary Fig. 1h). The coronal gyrus overlaps the anterior arm of the ectosylvian gyrus to a lesser degree than observed in extant *Lutra lutra* (Supplementary Fig. 4). The sigmoid gyri are more expanded than those of *Potamotherium*, but less expanded than those of modern otters.

*Promartes* (Supplementary Fig. 1g), *Pachycynodon* and *Phoberogale* have a narrow coronal gyrus). *Pachycynodon* lacks a cruciate sulcus and the sigmoid gyri are less expanded. *Phoberogale* and *Promartes* both have a clear cruciate sulcus. In all three taxa, the size and shape of the coronal gyrus is comparable to that of most modern terrestrial carnivorans (see below).

**Living carnivorans**. The coronal sulcus differs in shape, complexity, size and topographical orientation among modern taxa (Fig. 2a and representative species in Fig. 2b) . In most terrestrial carnivorans it is small and in the form of a simple arc (Fig. 2). The brain of the otter civet (*Cynogale bennettii*; Fig. 2b) has a clearly expanded coronal gyrus that is subdivided by a pair of secondary sulci (Supplementary Fig. 2a). Its size is nearly three times larger than that of other viverrids (Fig. 2a). The coronal gyrus is not of equal size in all otters. For example, the Eurasian

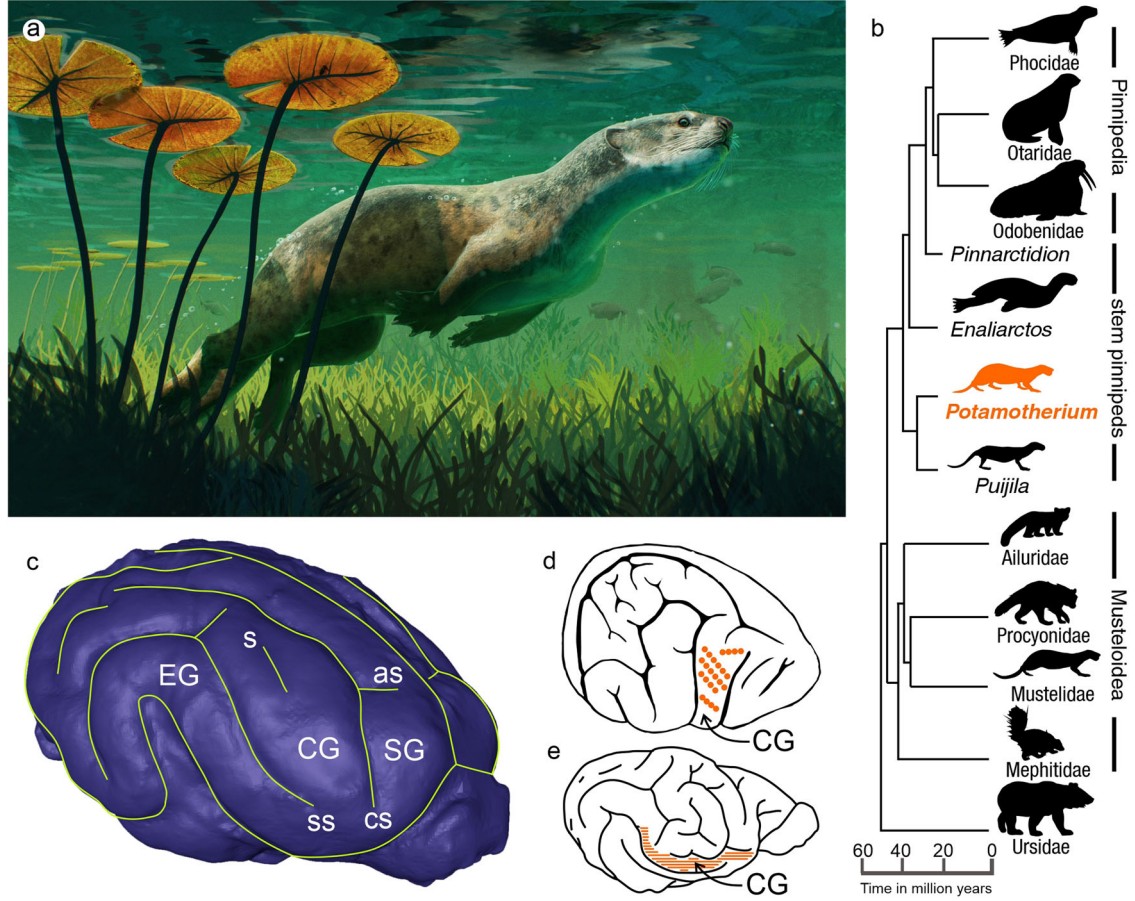

**Fig. 1 The phylogenetic position and brain of the early pinniped *Potamotherium* compared to other carnivorans. a** In vivo reconstruction of *Potamotherium valletoni* by Gabriel Ugueto. **b** Phyletic relationships among Arctoidea indicating the phylogenetic position of *Potamotherium* (phylogeny based on ref. [39]). **c** Digital representation of the *Potamotherium valletoni* endocast (FMNH PM 58906). ss suprasylvian sulcus, as ansate sulcus, cs coronal sulcus, s sulcus on the coronal gyrus, SG sigmoid gyrus, CG coronal gyrus, EG ectosylvian gyrus, Drawing by G.A. Lyras (**d**) Brain of the fur seal (*Callorhinus ursinus*). The series of points on the coronal gyrus (CG) marks the receptive field from the head (redrawn from ref. [20]). **e** Brain of the raccoon (*Procyon lotor*). The series of stripes indicates the extent of the representation of the head on the coronal gyrus (CG) (redrawn from ref. [25]). Images courtesy: Phocidae, Otariidae, *Enaliarctos*, *Potamotherium*, and Mustelidae by G.A. Lyras, Odobenidae, Procyonidae, and Mephitidae by Margot Michaud, *Puijila* by T. Michael Keesey, Ailuridae by Xavier Jenkins, Ursidae by Tracy Heath (CC0 1.0 Public Domain; https://www.phylopic.org/images).

otter (*Lutra lutra*; Fig. 2b), which lives in freshwater and has a mouth-oriented foraging behaviour, has a much-expanded coronal gyrus, bearing secondary sulci within it (Supplementary Figs. 1i and 2c). The coronal gyrus of the Asian small-clawed otter (*Aonyx cinerea*) is narrower with a dimple or short sulcus near its rostral end (Supplementary Fig. 2b).

The external surface area of the coronal gyrus of modern pinnipeds is larger than that of most terrestrially feeding carnivorans (Fig. 2c and Supplementary Table 1 for permutation tests and pairwise comparison). It should be noted, however, that it is not as enlarged as in *Cynogale*, *Lutra* or *Lontra* (Supplementary Fig. 2d). An interesting feature in living pinnipeds is the orientation of the coronal gyrus, which is nearly perpendicular to the ventral border of the brain. Furthermore, the sigmoid gyri are located extremely rostrally. Thus, the coronal gyrus is expanded in those modern taxa that primarily show mouth-oriented foraging behaviour (Fig. 2c). These taxa use their mystacial vibrissae to explore their environment.

**Reconstructed evolutionary history of the coronal gyrus.** Based on the above, the evolutionary history of the coronal gyrus in Carnivora can be outlined. The gyrus started as a narrow fold between the coronal and suprasylvian sulci. It remained narrow

in most terrestrial carnivorans (Fig. 1e). Most terrestrial carnivorans did not evolve any specialisation for enlarged cortical representations of peripheral sensory receptor fields from the head. In contrast, the coronal gyrus expanded disproportionately in semi-aquatic carnivorans that use their whiskers for exploring their environment. It did not expand in carnivorans that use their hands in foraging. The relative size of the coronal gyrus of modern pinnipeds is larger than that of most terrestrial carnivorans, but smaller than that of semiaquatic carnivorans that forage and capture prey using a mouth-oriented strategy (Fig. 2c).

Apart from changes in the relative size of the coronal gyrus, there have been notable changes in its topographic orientation. During the evolutionary history of stem pinnipeds, the coronal gyrus moved towards a more vertical orientation. The orientation of the coronal gyrus in *Potamotherium* is similar to that of terrestrial carnivorans. It is steeper in *Pinnarctidion* and even more vertical in modern seals (Fig. 3). This vertical orientation of the coronal gyrus is the arrangement seen in all modern pinnipeds.

**Discussion**

Radinsky[11] noticed several features in *Potamotherium* that are not seen in otters, such as a narrow sigmoid gyrus and a very broad coronal gyrus. He even went so far as to suggest that

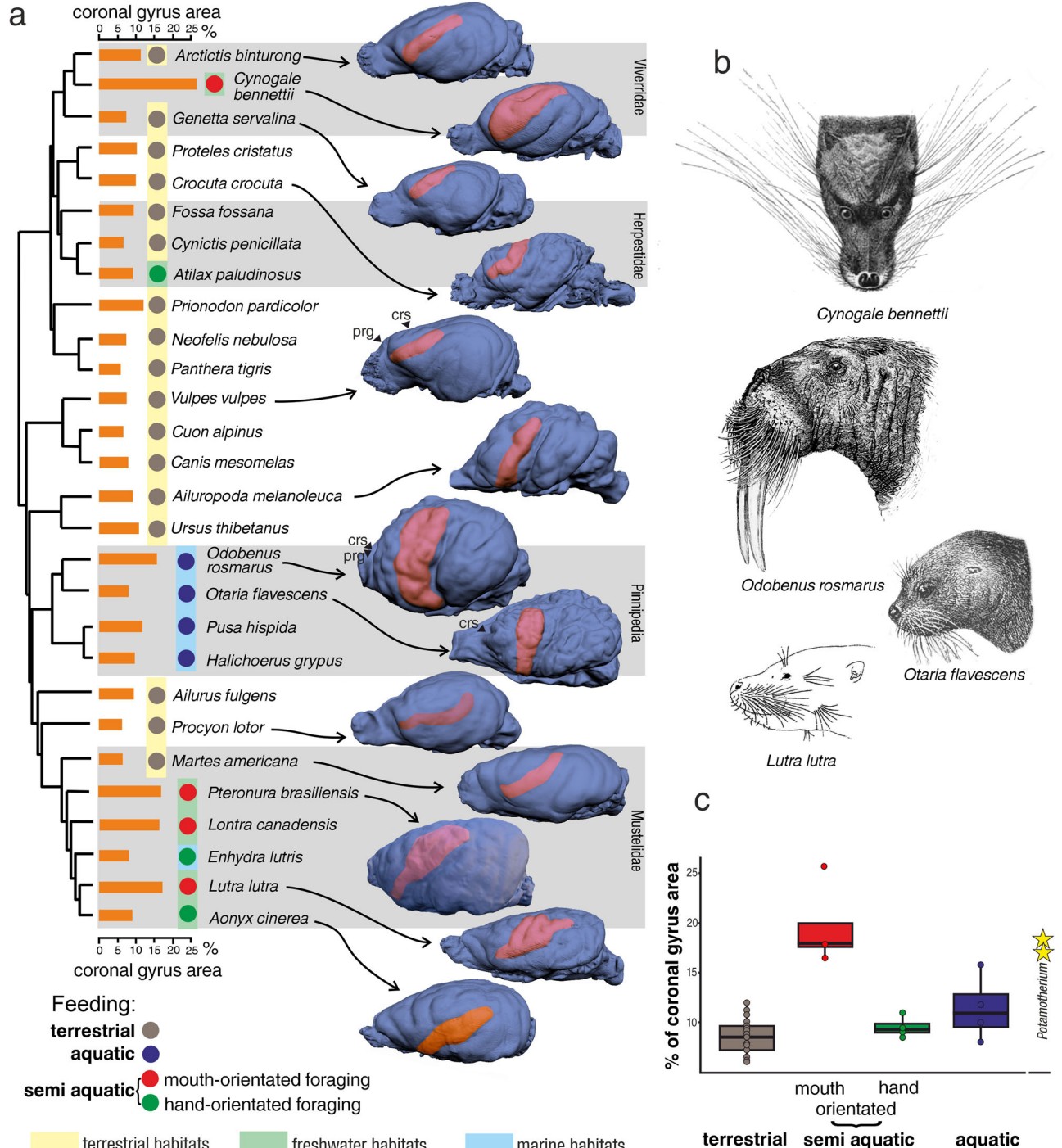

**Fig. 2 External brain morphology and the size of the coronal gyrus of living carnivorans. a** Phylogeny of selected modern carnivorans and the percentage of the superficially exposed surface area of the coronal gyrus compared to the total area of superficially exposed cerebral cortex. The coronal gyrus is marked on the endocasts in orange. The different feeding behaviours are indicated by coloured circles. The habitats are indicated by coloured stripes. In semiaquatic feeding, the prey is captured underwater but processed in the air, in contrast to aquatic feeding where the prey is captured and processed predominantly underwater. Carnivorans that exhibit semiaquatic feeding may forage and capture prey using either a mouth-oriented or hand-oriented strategy. The phylogeny is inferred from mitochondrial genomes[77]. prg: proreal gyrus; crs: cruciate gyrus. Endocasts are not to scale. See Supplementary Fig. 3 for renderings of endocasts not depicted here. Drawings by G.A. Lyras. **b** Examples of extant carnivoran taxa with developed facial vibrissae. **c** Boxplots of the relative size of the coronal gyrus among carnivorans with different feeding ecologies: terrestrial ($n = 19$ specimens), semiaquatic (where prey is captured underwater but processed out of the water) mouth-orientated ($n = 4$ specimens) and hand-orientated ($n = 4$ specimens), and aquatic or mostly aquatic feeding ($n = 4$ specimens). The midline represents the median. Asterisks indicate the position of the two *Potamotherium* specimens used in our analysis. See Supplementary Fig. 4 for a boxplot diagram annotated with the taxa used and Supplementary Table 2 for results from the permutation test and pairwise comparisons. Significant differences are found between mouth-orientated semi-aquatic versus terrestrial ($p = 0.0002$), terrestrial versus aquatic ($p = 0.0048$), and hand-orientated versus mouth-orientated ($p = 0.0048$). Line drawings credits: *Cynogale*[78], *Odobenus*[79], *Otaria* adapted from[79], and *Lutra*[80].

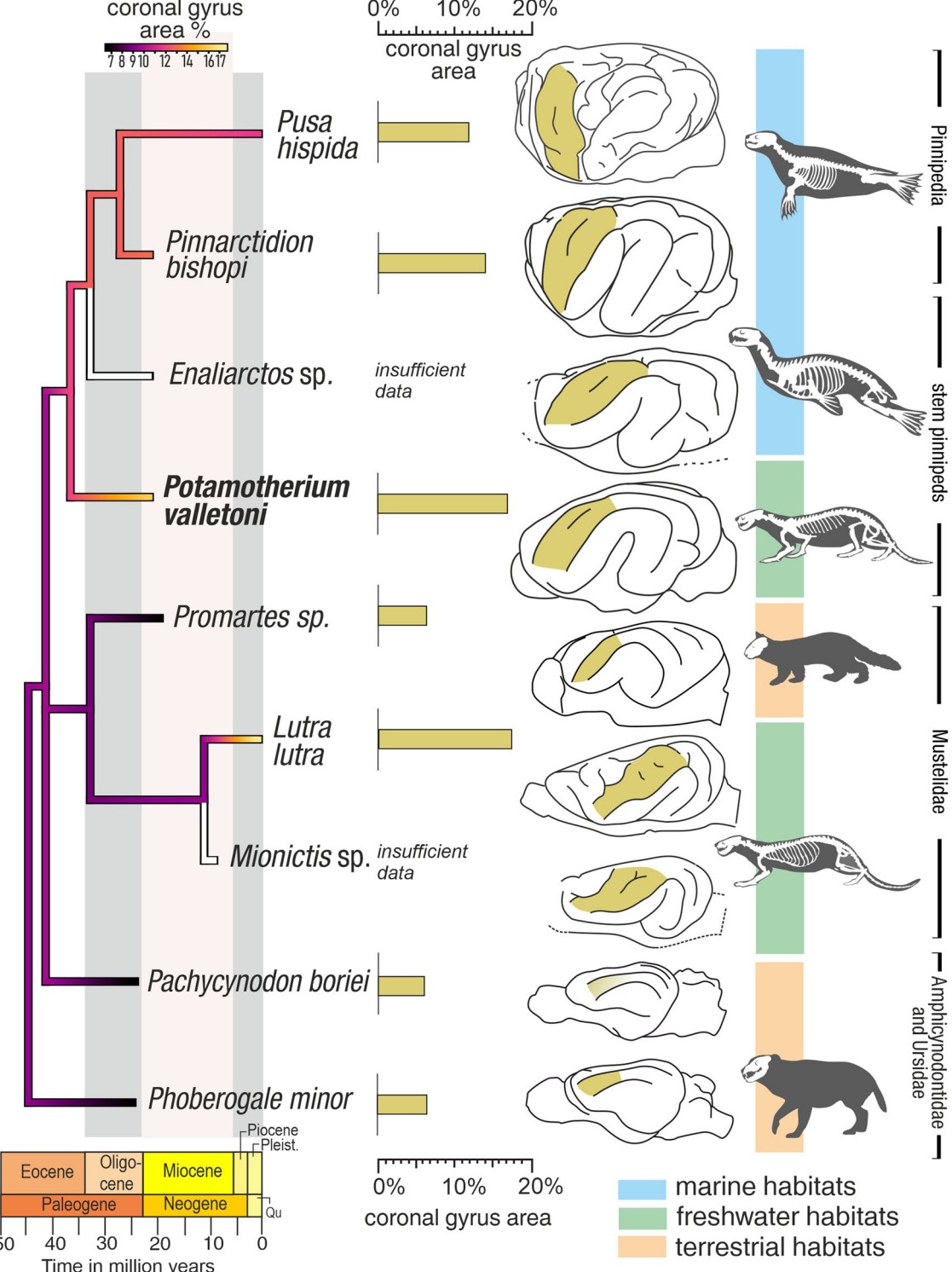

**Fig. 3 Phylogeny of selected fossil and living carnivorans and the percentage of superficially exposed surface area of the coronal gyrus, compared to the total area of superficially exposed cerebral cortex.** *Pachycynodon* is redrawn from ref. [28]. The phylogeny is based on a total evidence analysis[39]. The position of *Pachycynodon* is based on a phylogeny of basal arctoids[52]. The ancestral state reconstruction diagram was performed using refs. [73,74], and manually edited to accommodate for the missing data (the unedited diagram is available as Supplementary Fig. 5). The coronal gyrus is marked on the endocasts in orange. Endocasts are not to scale. Image courtesy: *Pusa, Enaliarctos, Potamotherium, Mionictis* and all endocasts by G.A. Lyras, *Promartes* adapted from image *Martes americana* by Gabriela Palomo-Muñoz, *Phoberogale* adapted from image *Cephalogale sharer* by T. Michael Keesey (CCO 1.0 Public Domain (www.phylopic.com).

*Potamotherium* could be related to seals rather than to otters. The phylogenetic position of *Potamotherium* has nevertheless remained controversial. It has been described as a mustelid[9,32–35], or as an early relative of Pinnipedia[8,36–39]. Our analysis adds further weight to the latter position, indicating that *Potamotherium* is indeed related to pinnipeds. The brain anatomy of *Potamotherium* has many similarities with *Enaliarctos* and *Pinnarctidion*, which are both widely accepted as pinnipeds.

Apart from its phylogenetic value, the brain of *Potamotherium* can inform us about its potential foraging behaviour. Electrophysiological mapping of the coronal gyrus in canids, felids, procyonids (Fig. 1e) and seals (Fig. 1d) shows that this region receives somatosensory input from the head[20,23–26]. The unequal expansion of the coronal gyrus in some modern carnivorans (e.g., *Cynogale*, *Lontra*) indicates enlarged cortical representations of peripheral sensory receptor fields from the head, and particularly from the vibrissae. These carnivorans use their mystacial vibrissae to explore their environment. The expanded coronal gyrus in the otter civet (*Cynogale bennetti*) and those otters that have a piscivorous mouth-oriented foraging behaviour, is also suggestive of increased facial tactile sensitivity[11,40,41]. Living pinnipeds have distinctive mystacial vibrissae, which they use to explore their environment. Hydrodynamic trail following has only been shown in the harbour seal (*Phoca vitulina*) and sea lion (*Zalophus californianus*)[42–44]. Sea lions, seals and walruses have been shown to be able to use their whiskers to differentiate objects by their size and shape, and to perform complex sensorimotor tasks[6,42,45].

Our results show an increased size of the coronal gyrus in *Potamotherium*, which provides further evidence that it was a whisker specialist. This confirms the hypothesis of increased tactile performance of mystacial vibrissae in stem pinnipeds as inferred from their enlarged infraorbital foramen[8].

An important note should be made here. The ability to sense with whiskers, instead of with hands, does not mean that *Potamotherium* exclusively relied on its whiskers. *Potamotherium* could have used its forelimbs for manipulating its prey. Modern carnivorans such as otters and seals, do so and the same has been suggested for *Enaliarctos*[3]. However, the ability to grasp and manipulate objects is more likely to involve specialisations of motor rather than somatic sensory cortex.

Based on the above, we postulate that the increased tactile performance of mystacial vibrissae of modern pinnipeds was already present around the beginning of the transition from a terrestrial to an aquatic lifestyle and facilitated their transition to an amphibious lifestyle (Fig. 1a).

## Methods

**Species data.** Our analysis includes endocranial casts from three stem pinnipeds (*Potamotherium*, *Enaliarctos* and *Pinnarctidion*), an early lutrine mustelid (*Mionictis*), an oligobunine mustelid (*Promartes*), an amphicynodontid (*Pachycynodon*), an archaic ursid (*Phoberogale*) and 31 species of extant carnivorans. For a list of fossil specimens, see Supplementary Table 2, for living species, see Supplementary Data 1). Availability of endocranial casts necessarily limited our selection of fossil taxa to these seven carnivoran species.

**Fossil species.** The genus *Potamotherium* is known from the Oligo-Miocene of Europe[9,46,47] and the early Miocene of North America[48]. It was a semiaquatic taxon adapted to freshwater habitats. Opinions regarding the phylogenetic position of *Potamotherium* have varied. Original commentators[32,49] considered it to be an otter, specifically of the genus *Lutra*. Some early authors also pointed to affinities with Viverridae[50,51]. The view that *Potamotherium* was a mustelid has continued to the present day[9,33,34,52]. Nevertheless, beginning with Tedford[36] *Potamotherium* has increasingly been assigned a basal position on the stem lineage of Pinnipedia[8,37–39]. Endocranial casts of *Potamotherium* have been described and figured elsewhere[9,11,27–31,53,54]. Here we include an endocranial cast of *Potamotherium valletoni* from France, originally described by Radinsky[29].

*Enaliarctos* was a marine amphibious animal with flippers, known from the early Miocene of California and Oregon[55]. Here, we use a partial natural endocast of *Enaliarctos*, figured and described as a 'late descendant of *Potamotherium* in North America'[11]. We attribute this specimen to *Enaliarctos* sp., based on its similarity to the endocranial anatomy of *Enaliarctos mealsi* (LACM CIT 5303) as figured by Michell and Tedford[56].

*Pinnarctidion* is also known from the early Miocene of California and Oregon[55]. It is considered sister taxon to the crown group of Pinnipedia that includes the three living pinniped families Otariidae (eared seals), Phocidae (earless seals) and Otobenidae (walruses)[39,57]. The endocranial cast used here has been described under the name *Enaliarctos mealsi*[47] and as *Enaliarctos* sp[31]. Here, we follow Barnes[58] in attributing this specimen to *Pinnarctidion bishopi*.

*Mionictis* is an early Lutrinae from North America[59]. Here we use a partial endocranial cast reproduced from a partial skull from the Miocene (Clarendonian) of Texas, described by Radinsky[11].

*Promartes* is an oligobunine terrestrial mustelid from the late Oligocene - early Miocene of North America[33]. It is contemporary with the stem pinnipeds that are examined here. The endocast used here has been described by Radinsky[29] and Mödden and Wolsan[30].

*Pachycynodon* is an arctoid carnivoran belonging to a basal ursoid group, generally referred to as Amphicynodontinae, which gave rise to Pinnipedimorpha[60]. Most of the known specimens of *Pachycynodon* are from the Oligocene of Europe[47]. *Phoberogale* is an early stem ursid. The endocast of *Phoberogale minor* used here has been described by Radinsky[29].

**Region of interest.** Electrophysiological mapping of the coronal gyrus in canids, felids, procyonids and seals (Fig. 1d, e) shows that this region receives somatosensory input from the head[20,23–26]. Because our aim is to reconstruct the importance of the whiskers in foraging behaviour in basal pinnipeds relative to that observed in crown pinnipeds on the one hand, and mustelids on the other, our region of interest here is the coronal gyrus. The description of endocasts will thus primarily focus on the representation of this gyrus.

**Data acquisition and analysis.** Endocasts were scanned with a Next Engine 3D laser scanner (Supplementary Table 2 for fossil species and Supplementary Data 1 for living species). The acquired scans were converted into closed mesh models and imported into Blender version 3.1.0 for surface area measurements. Blender is an open-source 3D computer graphics software used in creative arts and for scientific analysis and visualisation[61,62]. Endocasts were latex and plaster endocasts from made from skulls or fossil natural endocasts. All specimens are curated at natural history museum collections, as specified in Supplementary Table 1 and Supplementary Data 1.

For surface area calculation of the entire neocortex, we separately mark and measure the area bordered ventrally by the rhinal fissure and rostrally by the entry of the olfactory tract[63]. Since a proper distinction of the neocortex should be based on histological evidence, which is lacking in endocasts, we use the more general term 'cortex' instead of 'neocortex' throughout this paper. To calculate the surface area of the coronal gyrus (for extant and fossil taxa), we isolated the coronal gyri using sulcal maps from published works on Otariidae[21,64], Phocidae[20] and Odobenidae[65], Enaliarctidae[56], terrestrial Mustelidae[66,67], Lutrinae[11], Procyonidae[25], Canidae[68], Felidae[69], Ursidae[70], Viverridae and Herpestidae[40].

**Statistics and reproducibility.** For the ancestral state reconstruction and the statistical analysis, we used the package *phytools*[71] of the statistical software R[72] and the R code as developed by refs.[73,74]. To differentiate between groups (terrestrial, semi-aquatic and aquatic, where semi-aquatic species were further distinguished between mouth-orientated and hand-orientated feeding behaviours, we used a Fisher–Pitman permutation test, pairwise comparisons with significance level $p = 0.05$. For this, we used the function 'oneway_test' and 'pairwisePermutationTest' in the package *coin* (version 1.3.1) and *rcompanion* (version 2.3.25), respectively, to perform our permutation tests and the function 'ggboxplot' from the package *ggplot2* (version 3.3.0)[75] to graphically represent our results following R code as developed by refs.[73,74]. Surface area of the coronal gyrus was measured on 3D virtual models of the brain of 7 fossil and 31 extant carnivoran species ($n = 1$ per species, except for *Potamotherium*, $n = 2$).

**Reporting summary.** Further information on research design is available in the Nature Portfolio Reporting Summary linked to this article.

## Data availability

All data are available in the Supporting Information and Supplementary Data 1. Surface scans of endocasts generated for this project are available in MorphoSource[76] under project https://www.morphosource.org/projects/000532553.

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

## Acknowledgements

We thank Larry Heaney, William Simpson, and the late Bill Stanley (Field Museum of Natural History, Chicago, USA), John Flynn, Jin Meng and Judy Galkin (American Museum of Natural History, New York, USA), Christine Argot, Christine Lefèvre and Géraldine Veron (Muséum National d'Histoire Naturelle, Paris, France), and Loic Costeur (Naturhistorisches Museum, Basel, Switzerland) for providing access to collections under their charge. We thank Ursula Göhlich (Naturhistorisches Museum Wien, Austria), Edward Davis and Amanda Peng (Museum of Natural and Cultural History, University of Oregon, Eugene, USA) and Joanna Northover for providing us with data and photographs of specimens. G.L. received support from the SYNTHESYS Project (FR-TAF Call3 062).

## Author contributions

G.L. designed the study and made the figures. G.L. performed the analysis with input from A.G. G.L. led the writing with input from A.G. and L.W. Species data were provided by B.G. All authors read and approved the final manuscript.

## Competing interests

The authors declare no competing interests.
