## [Peer Review File · Communications Biology]

Reviewers' comments:

Reviewer #1 (Remarks to the Author):

Review of Underwater feeding of early seals: evidence from fossil brains

Thank you for the opportunity to review this article. This is a very interesting study on the evolution of brain in fossil seals. I particularly liked the ecological categories that go beyond "diet" and think more about behavioural ecology with the way animals forage for their food, which may have a direct impact on the size of brain regions that might be associated with a particular behaviour. The paper is overall well-written and easy to follow. I like the idea of quantifying the coronal gyrus, the inclusion of fossils to understand the ancestral condition and the analyse of the relation to sensory and behavioural ecology. The figures have nice illustrations, and the tables show the actual data that were gathered. However, I think that the manuscript could be improve in different ways described below. My review is relatively long, but I believe that those comments and suggestions would make the paper stronger.

The major aspects that should be addressed include the annotation of Figs. 2, and 3. I would suggest merging both figures into one single figure so Fig. S1 can be put in the main document. Statistical tests should be conducted for the boxplot (Fig. S1) and fossils should be included in this graph to see where they positioned compared to the modern sample.

Also, the surface of all endocasts should be stored into a repository such as Morphosource. <https://www.morphosource.org/>. The download settings can be set as "restricted download" so you can control how you share the data. It is important so they are not lost and for the reproducibility of the analyses.

Lastly, because there is a quantification of the surface of the coronal gyrus, I would suggest the authors to make an ancestral state reconstruction using the phylogeny, including both modern and fossil species (a new figure). R is a great tool for that specific analysis. It would be much more powerful at showing how the coronal gyrus changed through geological times within carnivora and in relation to feeding behaviour. The code to generate such figure is published and available in Github in the two following papers. Other studies have done this type of analyses but don't include the code.

<https://github.com/Bertrand-Ornella?tab=repositories>

Bertrand OC, Püschel HP, Schwab JA, Silcox MT, Brusatte SL (2021) The impact of locomotion on the brain evolution of squirrels and close relatives. *Commun. Biol*, 4, 1-15.

Bertrand OC, Shelley SL, Williamson TE, et al. (2022) Brawn before brains in placental mammals after the end-Cretaceous extinction. *Science*, 376, 80-85.

p. 2, line 28: I am always a bit bother with the term "pre-adapted" and I would use "exaptation" instead. The presence of high reliance on whiskers in a non-fully amphibious species related to modern pinnipeds is showing that this behaviour was probably crucial for the colonisation of this new environment. But preadaptation would mean that the presence of this feature was selected by the marine habitat. It is more likely that they had this behaviour which emerged for other reasons (or just randomly), and it became useful with the transition to an amphibious lifestyle. Triques and Christoffersen (2018) explain it well: "Exaptation is a trait that evolved for uses other than the current function or with no function at all and was later co-opted for its current function (Gomez-Mestre & Tejedo, 2005)".

Triques ML, Christoffersen ML (2018) Arguments for replacing the concept of preadaptation by

exaptation at the origin of terrestriality in Vertebrata. *Biological Journal of the Linnean Society*, 123, 235-246.

p. 2, line 59 to p. 3, line 64: I understand what the authors are explaining about the two different foraging strategies in pinnipeds, but the examples are disjointed. I would add "in the closely related species *Lutra* and *Aonyx*" because otherwise, the reader who is not familiar with the phylogenetic relationships of pinnipeds might be confused of why this comparison was made (same for the other example). Here, the goal is to show that there are convergences in behaviours within pinnipeds, and we are not sure what is the plesiomorphic condition for the clade. I would clarify this aspect because it is indeed an important point to make in the introduction.

p. 5, lines 79-81: I would delete "indicates that whiskers may have played an important role in their behaviour" and flesh out this section a little bit. The fact that stem pinnipeds have enlarged infraorbital foramina means that they would potentially have had thicker infraorbital nerves (based upon findings in modern species). This in turn would suggest that they had the same whisker-foraging behaviour as modern pinnipeds have today.

p. 5, line 88: I would say "many" instead of "all" because there might some details that are not preserved especially around the olfactory bulbs in mammals unless carnivores do indeed have brains that are completely against the endocranial cavity and differ from other groups.

p. 5, line 92: Could you add "of the neocortex" in front of gyrus to be more specific about which part of the brain you are focusing on? Somewhere else is fine too, I think it is important to say that it is about mapping the neocortex and not the whole brain.

p. 6, line 106: Because you are mentioning "terrestrial" in the text, in the legend of Fig. 2, please specify that all of the other species are "terrestrial" because you have "freshwater" and "marine", but nothing for the remaining taxa that don't belong to this category.

p. 6, lines 107-109: I would suggest adding an arrow in Fig. 2, showing the sulci (secondary fissures) that you are describing in *Cynogale*, it would help the reader that is not familiar with the topic.

p. 6, line 109: I would add in parentheses the actual values for the surface of coronal gyrus of *Cynogale* and other viverrids.

p. 6, lines 110-113: This section needs to have a reference to Fig. 2 as for *Cynogale*, I would add annotation on Fig. 2 to show exactly what is being described in the text for *Lutra* and *Aonyx*. Also add next to *Lutra* "mouth-oriented living in freshwater".

p. 6, lines 114-115: Is this result significant? (i.e., enlargement of the coronal gyrus in modern pinnipeds compared to terrestrial taxa). There is quite a bit of overlap. Please add statistics like pairwise comparison tests and add to Fig. S1 (and text where needed). For an example and code for the test to run, please see: Bertrand et al. (2021) "Permutation test and boxplots" in the methods.

Bertrand OC, Püschel HP, Schwab JA, Silcox MT, Brusatte SL (2021) The impact of locomotion on the brain evolution of squirrels and close relatives. *Commun. Biol*, 4, 1-15.

p. 6, line 115: The Figure S1 is a useful nice figure that encapsulate well the results. It would have been nice to have it in the main document. See my comment below for a solution to remain at 3 figures in the main document.

p. 6, line 116: I think it would be useful for the reader to add the major clades such as "pinnipeds" in Fig. 2 so they can localise where in the figure they have to look.

p. 6, lines 118-119: Again here, I would recommend the add some annotations to Fig. 2 on the endocasts so the reader can see where the "sigmoid gyri" and the "prorean gyrus" are on the specimens you are mentioning in the text.

p. 6, line 119: Change "ventrad" to "ventral"

p. 6, lines 119-120: Which taxa are "semi-aquatic" in Fig. 2? There is no legend for this particular category. Do you mean "freshwater". Please make sure that you clarify this aspect in the text and in the figure. You need to choose the terms and be consistent. Because it is about feeding, the categories in Table S1 should be the same in your figures 2 and S1, otherwise it is not easy to follow an brings confusion.

Fig. S1: If I understand correctly, you have 3 mouth-oriented semiaquatic taxa (Fig. 2) but in Table S1, I see 4 total. I think that there is an error in Fig. 2, I believe that *Enhydra lutris* should be "freshwater" (semiaquatic) right? Right now, it is labelled as "marine".

For the hand-oriented semiaquatic taxa (there is one missing on the figure, only 2/3 shown). Because your sample is relatively small, the case being made here should be strong, so adding *Pteronura brasiliensis* to Fig. 2 would help show the point that hand-oriented and mouth oriented feeders greatly differ not only in morphology but also in surface area of the coronal gyrus.

p. 7, Fig. 2: I would recommend adding a supplementary figure with all taxa with their respective endocast and coronal gyrus illustrated and not just selected species to show the morphological differences between the different categories as this is an important distinction between feeding behaviours.

p. 8, lines 128-134: Please make sure you refer to figures or tables within the results section when needed. Here, please cite Figure 1.

p. 8, line 129: it would be good to have some sort of quantification here on the fact that *Potamotherium* has a much wider coronal gyrus compared to modern terrestrial taxa of the same size.

p. 8, line 131: Add in the text next to "secondary sulcus" that you are referring to "sulcus on the coronal gyrus (s) in Figure 1".

p. 8, line 132: "compressed ventrally" compared to which taxa? Could you please clarify why and how the suprasylvian sulcus looks compressed?

p. 8, lines 127-153: In the "Fossil carnivores" section, please refer to Fig. 3 and other 2 when needed so it is easier to follow the description of the fossil taxa. The illustrations from Fig. 3 are nice, but they need to be annotated.

p. 8, lines 142-143: Referring to *Pinnarctidion*: "Just as in *Enaliarctos*, the rostral end of the brain is relatively high and caudally compressed" I am having difficulty seeing this similarity between both taxa when looking at the figure. Please clarify this sentence.

p. 8, line 147-148: Please cite a figure for this statement about the comparison with modern otters.

Fig. 3: I am not sure I understand why Figs. 2 and Fig. 3 are not together. Also, why having a selected sample instead of having all of the taxa in the figure? I would suggest merging Figs. 2 and 3 and, in that way, you can have Fig. S1 in the main document.

p. 9, line 155-173: The section of the reconstructed evolutionary history of the coronal gyrus does not have any figure cited. It is not clear if it belongs in the results or in the discussion as this stage. In order to really see the "ancestral state of the coronal gyrus" in carnivora, it would help to have both modern and extant taxa in the same figure with the phylogeny. Right now, it feels disjointed. A solution would be to make a figure of the ancestral state reconstruction using R.

p. 9, lines 163-164: Please rephrase this sentence: "The external surface area of the coronal gyrus of modern pinnipeds is larger than that of most terrestrial carnivores, but smaller than that of pinnipeds." You are comparing pinnipeds with themselves here, please clarify this sentence.

p. 11, line 191: Replace "Apart of" with "Apart from".

p. 12, line 218: Again, I would avoid the word "pre-adapted"

Fig. S1: I think it would be a good idea to add the fossils to the boxplot to see where they positioned compared to the modern sample.

p. 12, line 223: The nature and origin of each fossil endocast is clearly explained, which I really appreciate. However, the same is not true for the modern species unless I misread the text. Please add in the text how the endocasts of extant carnivores were obtained (i.e., plaster, latex or CT data).

p. 12, lines 227-228: Why not CT scanning crania of fossils to obtain virtual endocasts if natural/plaster endocasts limit your selection of fossils?

p. 15, line 283: The term "segmented" is used specifically in the context of 2D slices issued from data obtained through a micro-CT scanner, which is not the case here because the specimens were surface scanned here and not CT scanned. Also, you could not segment a surface. Please use another term such as "isolated" or "delimited" instead.

Also, please specify if this step was done on the extant sample. We need to know what was done to the modern taxa presented in Table S1. Even if this is the exact same treatment, it should be specified (as it should be for their origin and how they were obtained).

Table S1: Why using the term "Endocast number"? Are they not the collection numbers of the specimens? I would use the later to avoid confusion.

Table S2: Please add the age for each fossil and not just the locality. I know that it is in the text, but it would be nice to have it in the table for easy access.

Yours sincerely,
Ornella Bertrand

Reviewer #2 (Remarks to the Author):

This manuscript is well-written and the project is well conceived. I have only a few suggestions to the authors to improve this work.

Comments:

1. Page 5, line 87: "carnivores" should be "carnivorans"
2. Page 5, lines 87-88: Stating that "all details of the external brain anatomy" of carnivorans is preserved on endocasts is an overstatement. Several features of the ventral side of the brain are obscured on the endocast by the tracts and spaces occupied by various cranial nerves, and the hypophyseal cavity among other structures. Similarly, the dorsal surface of the midbrain (tectum) is covered by the cerebral hemispheres. I recommend that the authors instead say that "many details of the external brain anatomy..." or "many more details of the external brain anatomy are seen on carnivoran endocasts compared to endocasts of other mammalian groups..."
3. Page 6, line 115: "carnivores" should be "carnivorans"
4. Page 8, line 127: "carnivores" should be "carnivorans." This should be changed throughout the manuscript.
5. Page 8, line 145: the term "opercularized" should be defined or further explained here

Reviewer #3 (Remarks to the Author):

Understanding the temporal course of the evolution of sensory specializations is an interesting topic. The authors have specifically addressed whether ancestral pinnipeds potentially have highly sensitive vibrissae that may have been used to detect water-borne waves created by potential prey items.

In general I have no major concerns with this study, the delineation of the somatic sensory cortex across species, and the inference of the information from extant carnivores in terms of extinct species and potential behavioral correlates.

There are a few minor things that I'd really like the authors to address, as I believe to leave these unaddressed weakens the impact of the study.

In the abstract and elsewhere, the term "pre-adapted" or derivatives is used. I think this is used incorrectly. The Oxford dictionary defines preadaptation as: "an adaptation which serves a different purpose from the one for which it evolved". I don't believe this is the sense in which the authors are using this word. The authors are using pre-adapted in a deterministic sense, and this is not appropriate in evolutionary discussions. How about saying that the ancestral features may have proven advantageous in the transition from land to semi-aquatic environments? This takes away the deterministic inference in the term pre-adapted.

Hyperbole (meaning: exaggerated statements or claims not meant to be taken literally). The use of hyperbole is becoming entrenched in the modern scientific literature and it really shouldn't. To quote Donald Trump: "People want to believe that something is the biggest and the greatest and the most spectacular. I call it truthful hyperbole. It's an innocent form of exaggeration, and a very effective form of promotion". Unfortunately, the authors misuse hyperbole several times in their manuscript.

Abstract: "one of the most impressive ecological transitions", why not say simply say "significant"?
Line 40 "impressive" again. Line 100: "greatest". Line 108 "greatly". Line 109: "nearly three times greater". Line 111 "greatly". The discussion has a far more appropriate tone for a scientific paper. I ask the authors to understand that this paper will be published in a scientific journal, not a tweet, so hyperbole has no place in the scientific literature.

Line 30: "fully amphibious" why use "fully"?

Line 48: change "during their evolution" to "during pinniped evolution"

Line 50: change "pinnipeds fill the" to "pinnipeds occupied the"

Line 94: "development", don't you mean "the relative size"?

Lines 163/164: The sentence beginning: "The external surface ..." doesn't make any sense. Please reword or remove.

Line 206: change "increased size" to "increased relative size"

We thank the reviewers for their thorough review and insightful comments and suggestions for improvement. We adapted the text accordingly and provide detailed explanation per reviewer and per issue raised below.

Reviewer #1 (Remarks to the Author):

Thank you for the opportunity to review this article. This is a very interesting study on the evolution of brain in fossil seals. I particularly liked the ecological categories that go beyond “diet” and think more about behavioural ecology with the way animals forage for their food, which may have a direct impact on the size of brain regions that might be associated with a particular behaviour. The paper is overall well-written and easy to follow. I like the idea of quantifying the coronal gyrus, the inclusion of fossils to understand the ancestral condition and the analyse of the relation to sensory and behavioural ecology. The figures have nice illustrations, and the tables show the actual data that were gathered. However, I think that the manuscript could be improve in different ways described below. My review is relatively long, but I believe that those comments and suggestions would make the paper stronger.

The major aspects that should be addressed include the annotation of Figs. 2, and 3. I would suggest merging both figures into one single figure so Fig. S1 can be put in the main document. Statistical tests should be conducted for the boxplot (Fig. S1) and fossils should be included in this graph to see where they positioned compared to the modern sample.

Response: We agree with the reviewer, and now include the boxplot (originally included as Supplementary Figure 1) within Figure 2 of the main text. The new Fig. 2 consists of two parts: (a) is a modified version of the original Fig.2 and (b) is the inserted boxplot. Also, we added *Potamotherium* to the boxplot. In addition, as requested, we ran additional statistical tests and added a new supplementary table with pairwise comparison tests (Supplementary Table 3).

In addition, we now added new supplementary figures depicting anatomical details and aspects of the endocasts not shown in the three figures of the main document.

However, we declined to merge Figs. 2 and Fig. 3, but prefer to maintain the separation. The aim of Fig. 2 is to summarise the diversity of coronal gyrus configuration in modern carnivores and to graphically demonstrate its ecological significance. Once the connection between coronal gyrus and foraging behaviour is established, the reader can move on to Fig.3. This figure provides an overview of the evolution of the coronal gyrus in pinnipeds and related arctoids.

We explain this and other points in more detail below.

Also, the surface of all endocasts should be stored into a repository such as Morphosource. <https://www.morphosource.org/>. The download settings can be set as “restricted download” so you can control how you share the data. It is important so they are not lost and for the reproducibility of the analyses.

Response: We support the use of Morphosource. We already contacted museum curators and asked permission to upload surface-scans of specimens under their charge. We will do so in the coming weeks.

Lastly, because there is a quantification of the surface of the coronal gyrus, I would suggest the authors to make an ancestral state reconstruction using the phylogeny, including both modern and fossil species (a new figure). R is a great tool for that specific analysis. It would be much more powerful at showing how the coronal gyrus changed through geological times within carnivora and

in relation to feeding behaviour. The code to generate such figure is published and available in Github in the two following papers. Other studies have done this type of analyses but don't include the code.

<https://github.com/Bertrand-Ornella?tab=repositories>

Bertrand OC, Püschel HP, Schwab JA, Silcox MT, Brusatte SL (2021) The impact of locomotion on the brain evolution of squirrels and close relatives. *Commun. Biol*, 4, 1-15.

Bertrand OC, Shelley SL, Williamson TE, et al. (2022) Brawn before brains in placental mammals after the end-Cretaceous extinction. *Science*, 376, 80-85.

Response: In order to build the boxplots, run the statistical tests and create the ancestral state reconstruction diagrams we used the R code that was indicated by the reviewer. We now also refer to the literature indicated by the reviewer. The ancestral state reconstruction diagram was added to Fig. 3 (see above). Because the diagram was manually edited to accommodate *Enaliarctos* and *Mionictis* (for which data are lacking), we placed an unedited diagram in a new Supplementary Fig. 5.

p. 2, line 28: I am always a bit bother with the term “pre-adapted” and I would use “exaptation” instead. The presence of high reliance on whiskers in a non-fully amphibious species related to modern pinnipeds is showing that this behaviour was probably crucial for the colonisation of this new environment. But preadaptation would mean that the presence of this feature was selected by the marine habitat. It is more likely that they had this behaviour which emerged for other reasons (or just randomly), and it became useful with the transition to an amphibious lifestyle. Triques and Christoffersen (2018) explain it well: “Exaptation is a trait that evolved for uses other than the current function or with no function at all and was later co-opted for its current function (Gomez-Mestre & Tejedo, 2005)”.

Triques ML, Christoffersen ML (2018) Arguments for replacing the concept of preadaptation by exaptation at the origin of terrestriality in Vertebrata. *Biological Journal of the Linnean Society*, 123, 235-246.

Response: the reviewer is absolutely right in how the terms “pre-adapted” and “exaptation” should be used. However, we choose to refrain from both terms, as we realize that neither is correct here. The whiskers are not used (essentially) differently in marine and freshwater habitats after all, so there is no shift in usage. We changed this sentence “.....is an ancestral feature that pre-adapted stem pinnipeds for marine habitats.” Into “.....is an ancestral feature that favoured survival of stem pinnipeds in marine habitats.”

p. 2, line 59 to p. 3, line 64: I understand what the authors are explaining about the two different foraging strategies in pinnipeds, but the examples are disjointed. I would add “in the closely related species *Lutra* and *Aonyx*” because otherwise, the reader who is not familiar with the phylogenetic relationships of pinnipeds might be confused of why this comparison was made (same for the other example). Here, the goal is to show that there are convergences in behaviours within pinnipeds, and we are not sure what is the plesiomorphic condition for the clade. I would clarify this aspect because it is indeed an important point to make in the introduction.

Response: indeed, this was a confusing part. We rephrased this part

p. 5, lines 79-81: I would delete “indicates that whiskers may have played an important role in their behaviour” and flesh out this section a little bit. The fact that stem pinnipeds have enlarged infraorbital foramina means that they would potentially have had thicker infraorbital nerves (based

upon findings in modern species). This in turn would suggest that they had the same whisker-foraging behaviour as modern pinnipeds have today.

Response: we politely refrain from this deletion. As we explain in the text, the size of infraorbital foramen alone cannot be used to predict whisker sensitivity, because its size depends not only on the number of whiskers, but also on the innervation of the individual whiskers (Muchlinski et al. 2020; Milne et al. 2022).

p. 5, line 88: I would say “many” instead of “all” because there might be some details that are not preserved especially around the olfactory bulbs in mammals unless carnivores do indeed have brains that are completely against the endocranial cavity and differ from other groups.

Response: indeed, this is correct; we followed this recommendation.

p. 5, line 92: Could you add “of the neocortex” in front of gyrus to be more specific about which part of the brain you are focusing on? Somewhere else is fine too, I think it is important to say that it is about mapping the neocortex and not the whole brain.

Response: added as suggested.

p. 6, line 106: Because you are mentioning “terrestrial” in the text, in the legend of Fig. 2, please specify that all of the other species are “terrestrial” because you have “freshwater” and “marine”, but nothing for the remaining taxa that don’t belong to this category.

Response: we modified Fig.2. Now the annotation of the figure is more clear. We indicate with different colours the habitat and the feeding/foraging strategy of each taxon.

p. 6, lines 107-109: I would suggest adding an arrow in Fig. 2, showing the sulci (secondary fissures) that you are describing in *Cynogale*, it would help the reader that is not familiar with the topic.

Response: Because figure Fig. 2 is very complex already, we created a new Supplementary Fig. 1 where we indicate the secondary fissures in *Cynogale*. In the same figure we included endocasts from three otters (*Aonyx*, *Lutra* and *Londra*), where we annotated the names of sulci mentioned in the text.

p. 6, line 109: I would add in parentheses the actual values for the surface of coronal gyrus of *Cynogale* and other viverrids.

Response: we added a citation to Fig.2.

p. 6, lines 110-113: This section needs to have a reference to Fig. 2 as for *Cynogale*, I would add annotation on Fig. 2 to show exactly what is being described in the text for *Lutra* and *Aonyx*. Also add next to *Lutra* “mouth-oriented living in freshwater”.

Response: we created a new Supplementary Fig. 1. In this figure we annotated the anatomical features mentioned in the text. We also added “mouth-oriented, living in freshwater” next to Eurasian otter (*Lutra lutra*) as suggested.

p. 6, lines 114-115: Is this result significant? (i.e., enlargement of the coronal gyrus in modern pinnipeds compared to terrestrial taxa). There is quite a bit of overlap. Please add statistics like pairwise comparison tests and add to Fig. S1 (and text where needed). For an example and code for the test to run, please see: Bertrand et al. (2021) “Permutation test and boxplots” in the methods.

Bertrand OC, Püschel HP, Schwab JA, Silcox MT, Brusatte SL (2021) The impact of locomotion on the brain evolution of squirrels and close relatives. *Commun. Biol.*, 4, 1-15.

Response: Using the R code indicated by the reviewer we created a new boxplot (now as Fig. 2c and as Supplementary Fig. 2) and we ran pairwise comparison tests (now in Supplementary Table 3). We added the suggested reference to the Methods section and Supplementary Table 3.

p. 6, line 115: The Figure S1 is a useful nice figure that encapsulate well the results. It would have been nice to have it in the main document. See my comment below for a solution to remain at 3 figures in the main document.

Response: We agree with the reviewer. Fig. S1 is now part of Fig. 2.

p. 6, line 116: I think it would be useful for the reader to add the major clades such as “pinnipeds” in Fig. 2 so they can localise where in the figure they have to look.

Response: We did so. Clades mentioned in the text are now indicated in the figure. We also shaded those clades so that they are easier to identify.

p. 6, lines 118-119: Again here, I would recommend the add some annotations to Fig. 2 on the endocasts so the reader can see where the “sigmoid gyri” and the “prorean gyrus” are on the specimens you are mentioning in the text.

Response: We added the suggested annotation in Fig.2. The position of prorean and sigmoid gyri is indicated for two pinnipeds and one terrestrial carnivoran (*Vulpes*) for comparison.

p. 6, line 119: Change “ventrad” to “ventral”

Response: not needed anymore, as we removed that line.

p. 6, lines 119-120: Which taxa are “semi-aquatic” in Fig. 2? There is no legend for this particular category. Do you mean “freshwater”. Please make sure that you clarify this aspect in the text and in the figure. You need to choose the terms and be consistent. Because it is about feeding, the categories in Table S1 should be the same in your figures 2 and S1, otherwise it is not easy to follow an brings confusion.

Response: we rephrased the legend for and adapted the caption of Fig. 2(a). Now it is clear that the term semi-aquatic refers to the feeding behaviour and not to the habitat. We thank the reviewer for pointing this out this confusing issue.

Fig. S1: If I understand correctly, you have 3 mouth-oriented semiaquatic taxa (Fig. 2) but in Table S1, I see 4 total. I think that there is an error in Fig. 2, I believe that *Enhydra lutris* should be “freshwater” (semiaquatic) right? Right now, it is labelled as “marine”.

Response: *Enhydra lutris* is a sea otter.

For the hand-oriented semiaquatic taxa (there is one missing on the figure, only 2/3 shown). Because your sample is relatively small, the case being made here should be strong, so adding *Pteronura brasiliensis* to Fig. 2 would help show the point that hand-oriented and mouth oriented feeders greatly differ not only in morphology but also in surface area of the coronal gyrus.

Response: we added *Pteronura brasiliensis* to Fig. 2a.

p. 7, Fig. 2: I would recommend adding a supplementary figure with all taxa with their respective endocast and coronal gyrus illustrated and not just selected species to show the morphological

differences between the different categories as this is an important distinction between feeding behaviours.

Response: we created a new Supplementary Fig. 2 where we depict endocasts from the taxa not shown in Fig. 2.

p. 8, lines 128-134: Please make sure you refer to figures or tables within the results section when needed. Here, please cite Figure 1.

Response: we added citations to figures and tables.

p. 8, line 129: it would be good to have some sort of quantification here on the fact that Potamotherium has a much wider coronal gyrus compared to modern terrestrial taxa of the same size.

Response: instead of measuring, we added a citation (Radinsky 1968).

p. 8, line 131: Add in the text next to “secondary sulcus” that you are referring to “sulcus on the coronal gyrus (s) in Figure 1”.

Response: we added a citation to Fig. 1c.

p. 8, line 132: “compressed ventrally” compared to which taxa? Could you please clarify why and how the suprasylvian sulcus looks compressed?

Response: we rephrased as follows: “the anterior limb of the suprasylvian sulcus is shifted ventrally”

p. 8, lines 127-153: In the “Fossil carnivores” section, please refer to Fig. 3 and other 2 when needed so it is easier to follow the description of the fossil taxa. The illustrations from Fig. 3 are nice, but they need to be annotated.

Response: the aim of Fig. 3 is to graphically illustrate the evolution of coronal gyrus. It does so by remaining simple. We nevertheless agree with the reviewer that more information should be provided to the readers. We therefore added a new Supplementary Fig. 4, which has detailed depictions of the stem pinniped endocasts.

p. 8, lines 142-143: Referring to Pinnarctidion: “Just as in Enaliarctos, the rostral end of the brain is relatively high and caudally compressed” I am having difficulty seeing this similarity between both taxa when looking at the figure. Please clarify this sentence.

Response: we understand the difficulty of seeing the differences between *Pinnarctidion* and *Enaliarctos*. Fig. 3 is meant only to demonstrate the evolution of the coronal gyrus and thus other details of their anatomy are not easily seen. We therefore created a new Supplementary Fig. 5, where detailed pictures of the stem pinniped endocasts are presented.

p. 8, line 147-148: Please cite a figure for this statement about the comparison with modern otters.

Response: we cited Fig. 3.

Fig. 3: I am not sure I understand why Figs. 2 and Fig. 3 are not together. Also, why having a selected sample instead of having all of the taxa in the figure? I would suggest merging Figs. 2 and 3 and, in that way, you can have Fig. S1 in the main document.

Response: Fig. 2 summarises the diversity of the coronal gyrus in modern carnivorans and graphically demonstrates its functional significance. Fig. 3 summarises the evolution of coronal gyrus in pinnipeds and related arctoids.

p. 9, line 155-173: The section of the reconstructed evolutionary history of the coronal gyrus does not have any figure cited. It is not clear if it belongs in the results or in the discussion as this stage. In order to really see the “ancestral state of the coronal gyrus” in carnivora, it would help to have both modern and extant taxa in the same figure with the phylogeny. Right now, it feels disjointed. A solution would be to make a figure of the ancestral state reconstruction using R.

Response: we used the R code developed by the reviewer and colleagues for building an ancestral state reconstruction diagram. The diagram was added to Fig. 3. Because the diagram was manually edited to accommodate *Enaliarctos* and *Mionictis* (for which we have no data) we placed the unedited diagram in a new Supplementary Fig. 5.

p. 9, lines 163-164: Please rephrase this sentence: “The external surface area of the coronal gyrus of modern pinnipeds is larger than that of most terrestrial carnivores, but smaller than that of pinnipeds.” You are comparing pinnipeds with themselves here, please clarify this sentence.

Response: we rephrased as follows: “The relative size of the coronal gyrus of modern pinnipeds is larger than that of most terrestrial carnivorans, but smaller than that of semiaquatic carnivorans that forage and capture prey using a mouth oriented strategy (Fig. 2c)”

p. 11, line 191: Replace “Apart of” with “Apart from”.

Response: repaired, thanks.

p. 12, line 218: Again, I would avoid the word “pre-adapted”

Response: we deleted pre-adapted, see also above

Fig. S1: I think it would be a good idea to add the fossils to the boxplot to see where they positioned compared to the modern sample.

Response: we added *Potamotherium* to the boxplot.

p. 12, line 223: The nature and origin of each fossil endocast is clearly explained, which I really appreciate. However, the same is not true for the modern species unless I misread the text. Please add in the text how the endocasts of extant carnivores were obtained (i.e., plaster, latex or CT data).

Response: we added the following part “We included in our analysis 31 endocasts from extant carnivores. These were latex and plaster endocasts from natural history museum collections (see Supplementary Table 1)”.

p. 12, lines 227-228: Why not CT scanning crania of fossils to obtain virtual endocasts if natural/plaster endocasts limit your selection of fossils?

Response: we do not have the means to do so, and frankly speaking, the results should be the same.

p. 15, line 283: The term “segmented” is used specifically in the context of 2D slices issued from data obtained through a micro-CT scanner, which is not the case here because the specimens were surface scanned here and not CT scanned. Also, you could not segment a surface. Please use another term such as “isolated” or “delimitated” instead.

Response: we now use the word “isolated”.

Also, please specify if this step was done on the extant sample. We need to know what was done to the modern taxa presented in Table S1. Even if this is the exact same treatment, it should be specified (as it should be for their origin and how they were obtained).

Response: we added that it was done both on extant and fossil taxa.

Table S1: Why using the term “Endocast number”? Are they not the collection numbers of the specimens? I would use the later to avoid confusion.

Response: we understand the confusion. We replaced the term “Endocast number” with the term “specimen accession number”. That is the museum catalogue number for each specimen.

Table S2: Please add the age for each fossil and not just the locality. I know that it is in the text, but it would be nice to have it in the table for easy access.

Response: we did so.

Yours sincerely,

Ornella Bertrand

Reviewer #2 (Remarks to the Author):

This manuscript is well-written and the project is well conceived. I have only a few suggestions to the authors to improve this work.

Comments:

1. Page 5, line 87: “carnivores” should be “carnivorans”

Response: We replaced the word “carnivores” with “carnivorans” (and did so for the other instances mentioned below)

2. Page 5, lines 87-88: Stating that “all details of the external brain anatomy” of carnivorans is preserved on endocasts is an overstatement. Several features of the ventral side of the brain are obscured on the endocast by the tracts and spaces occupied by various cranial nerves, and the hypophyseal cavity among other structures. Similarly, the dorsal surface of the midbrain (tectum) is covered by the cerebral hemispheres. I recommend that the authors instead say that “many details of the external brain anatomy...” or “many more details of the external brain anatomy are seen on carnivoran endocasts compared to endocasts of other mammalian groups...”

Response: we agree with the reviewer and thankful for pointing out this mistake. We replaced the word “all” with “most”.

3. Page 6, line 115: “carnivores” should be “carnivorans”

4. Page 8, line 127: “carnivores” should be “carnivorans.” This should be changed throughout the manuscript.

5. Page 8, line 145: the term “opercularized” should be defined or further explained here

Response: Instead of using this technical term, we rephrased as follows: “the coronal gyrus overlaps the anterior arm of the ectosylvian gyrus”. The comment of the reviewer made us realise that more detailed depictions of endocasts are needed. We therefore added depictions of mustelid endocasts

in the Supplementary Fig. 4 and we annotated them accordingly. This will help the specialist reader to understand the descriptions mentioned in the text.

Reviewer #3 (Remarks to the Author):

Understanding the temporal course of the evolution of sensory specializations is an interesting topic. The authors have specifically addressed whether ancestral pinnipeds potentially have highly sensitive vibrissae that may have been used to detect water-borne waves created by potential prey items.

In general I have no major concerns with this study, the delineation of the somatic sensory cortex across species, and the inference of the information from extant carnivores in terms of extinct species and potential behavioral correlates.

There are a few minor things that I'd really like the authors to address, as I believe to leave these unaddressed weakens the impact of the study.

In the abstract and elsewhere, the term "pre-adapted" or derivatives is used. I think this is used incorrectly. The Oxford dictionary defines preadaptation as: "an adaptation which serves a different purpose from the one for which it evolved". I don't believe this is the sense in which the authors are using this word. The authors are using pre-adapted in a deterministic sense, and this is not appropriate in evolutionary discussions. How about saying that the ancestral features may have proven advantageous in the transition from land to semi-aquatic environments? This takes away the deterministic inference in the term pre-adapted.

Response: this is a good point, and was also raised by Reviewer 1. We now follow the advice of Reviewer 3 and simply state that we think this ancestral feature was advantageous. In this way we avoid pre-adapted and exadaptation. See also under Reviewer 1.

Hyperbole (meaning: exaggerated statements or claims not meant to be taken literally). The use of hyperbole is becoming entrenched in the modern scientific literature and it really shouldn't. To quote Donald Trump: "People want to believe that something is the biggest and the greatest and the most spectacular. I call it truthful hyperbole. It's an innocent form of exaggeration, and a very effective form of promotion". Unfortunately, the authors misuse hyperbole several times in their manuscript. Abstract: "one of the most impressive ecological transitions", why not say simply say "significant"? Line 40 "impressive" again. Line 100: "greatest". Line 108 "greatly". Line 109: "nearly three times greater". Line 111 "greatly". The discussion has a far more appropriate tone for a scientific paper. I ask the authors to understand that this paper will be published in a scientific journal, not a tweet, so hyperbole has no place in the scientific literature.

Response: yes, the reviewer may have a point. We removed them.

Line 30: "fully amphibious" why use "fully"?

Response: correct, this adds nothing, we removed the word "fully".

Line 48: change "during their evolution" to "during pinniped evolution"

Response: changed as suggested.

Line 50: change "pinnipeds fill the" to "pinnipeds occupied the"

Response: changed as suggested.

Line 94: "development", don't you mean "the relative size"?

Response: indeed, we mean exactly that. We changed the word "development" with "the relative size". Now the meaning of the sentence is more accurate.

Lines 163/164: The sentence beginning: "The external surface ..." doesn't make any sense. Please reword or remove.

Response: we thank the reviewer for spotting this awkward mistake; that sentence made no sense as we were comparing pinnipeds with themselves. We corrected that line as follows: "The relative size of the coronal gyrus of modern pinnipeds is larger than that of most terrestrial carnivorans, but smaller than that of semiaquatic carnivorans that forage and capture prey using a mouth-oriented strategy (Fig. 2c)"

Line 206: change "increased size" to "increased relative size"

Response: changed as suggested.

REVIEWERS' COMMENTS:

Reviewer #1 (Remarks to the Author):

I thank the authors for incorporating my comments. The manuscript is much improved. I do not have any further comments except a minor suggestion.

Figure 3: Add a legend for the ancestral state reconstruction. (i.e., 6.4 to 17.4). I understand that this is the coronal gyrus area percentage, but you should add it as you did for the histogram next to it.

Your sincerely,

Ornella Bertrand

Reviewer #3 (Remarks to the Author):

The authors have made all changes suggested and the paper is in a very good state currently. No further suggestions from my perspective.